

# Intensity of bouted and sporadic physical activity and the metabolic syndrome in adults

Jordan Robson[1] and Ian Janssen[1,2]

[1] School of Kinesiology and Health Studies, Queen's University, Kingston, ON, Canada
[2] Department of Public Health Sciences, Queen's University, Kingston, ON, Canada

## ABSTRACT

**Background.** Physical activity guidelines for adults only recognize the health benefits of accumulating bouted moderate-to-vigorous physical activity (MVPA), or MVPA occurring over at least 10 consecutive minutes. There is a lack of evidence supporting the health benefits of other patterns and intensities of activity including sporadic MVPA (i.e., MVPA occurring in periods of fewer than 10 consecutive minutes) and light intensity physical activity (LIPA). The objective of this study was to examine the health benefits associated with physical activity that does not meet the physical activity guidelines criteria for bouted MVPA. Specifically, we examined the association between sporadic MVPA and bouted and sporadic LIPA with the metabolic syndrome.
**Methods.** We studied a representative cross-sectional sample of 1,974 adults aged 20 years and older from the 2003–2006 US National Health and Nutrition Examination Survey. Physical activity was measured over 7 days using Actigraph AM-7164 accelerometers. Each minute over the 7-day measurement period was classified as being of a sedentary, light, or moderate-to-vigorous intensity. A 10 min threshold differentiated bouted activity from sporadic activity. Average minutes/day of sporadic LIPA, sporadic MVPA, bouted LIPA, bouted MVPA, and embedded MVPA (MVPA occurring within bouts of primarily LIPA) were calculated. Metabolic syndrome status was determined using established criteria. Associations were examined using logistic regression and controlled for relevant covariates.
**Results.** For every 30 min/day of physical activity, the odds ratio (95% confidence interval) of the metabolic syndrome was reduced by 4% (1–7%) for bouted LIPA, 64% (51–71%) for bouted MVPA, and 57% (45–67%) for embedded MVPA. Sporadic LIPA was not independently associated with the metabolic syndrome. We could not examine the association between sporadic MVPA and the metabolic syndrome because participants accumulated such a marginal amount of this type of activity (i.e., median = 2 min/day, only 11% of participants accumulated ≥5 min/day).
**Conclusion.** The intensity of non-bouted activity is important, as embedded MVPA had a stronger association with the metabolic syndrome than sporadic LIPA and a comparable association to bouted MVPA.

Corresponding author
Ian Janssen, ian.janssen@queensu.ca

## INTRODUCTION

Physical activity is an important determinant of morbidity and mortality within adults (*Tremblay et al., 2011*; *Warburton, Nicol & Bredin, 2006*). Physical activity can be of a light, moderate, or vigorous intensity. The latter two intensities are often combined to form moderate-to-vigorous physical activity (MVPA) (*Tremblay et al., 2011*; *Warburton, Nicol & Bredin, 2006*). Physical activity can be accumulated in bouts or sporadically. A bout refers to a session of physical activity that lasts at least 10 consecutive minutes, while sporadic physical activity refers to activity that lasts for fewer than 10 min (*Clarke & Janssen, 2014*; *Glazer et al., 2013*; *Strath et al., 2008*). When the intensity and frequency of physical activity are considered simultaneously, several different patterns of physical activity emerge. These different patterns are named and described in Table 1. The overarching goal of this study was to examine how these different patterns of physical activity influence health.

Physical activity guidelines for adults only recognize the health benefits of bouted MVPA (*Physical Activity Guidelines Advisory Committee, 2008*; *Tremblay et al., 2011*; *World Health Organization, 2010*). The exclusion of sporadic MVPA implies that the there is a lack of evidence supporting the health benefits of this pattern of physical activity. However, recent findings indicate that sporadic MVPA is associated with cardiometabolic risk factors (*Clarke & Janssen, 2014*; *Glazer et al., 2013*; *Strath et al., 2008*).

There is conflicting evidence as to whether sporadic light intensity physical activity (LIPA) has health benefits. Two small experimental studies reported that breaking up prolonged periods of sedentary behavior with sporadic LIPA improved acute postprandial glucose response (*Bailey & Locke, 2015*; *Dunstan et al., 2012*). Conversely, two observational studies of adults with abdominal obesity who did not accumulate bouted MVPA reported that sporadic LIPA, as assessed by accelerometery over 7 days, was not associated with cardiorespiratory fitness or abdominal obesity (*McGuire & Ross, 2011*; *McGuire & Ross, 2012*).

Non-bouted MVPA can occur in a truly sporadic manner (*Robson & Janssen, 2015*). An example of this pattern of activity is a 3 min brisk walk that is preceded and followed by sitting. Non-bouted MVPA can also occur embedded within bouts of activity that are primarily comprised of LIPA (*Robson & Janssen, 2015*). An example of this pattern of activity is a 3 min brisk walk that is preceded and followed by slow walking (*Robson & Janssen, 2015*). We refer to these two forms of non-bouted MVPA as sporadic MVPA and embedded MVPA, respectively. A typical adult accumulates only 2 min/day of sporadic MVPA, but 16 min/day of embedded MVPA, the latter of which accounts for 85% of total daily MVPA (*Robson & Janssen, 2015*). Previous studies examining the health benefits of sporadic MVPA did not differentiate between these two forms of non-bouted MVPA (*Clarke & Janssen, 2014*; *Glazer et al., 2013*; *Strath et al., 2008*).

The objective of this study was to examine the health benefits associated with physical activity that does not meet the bouted MVPA criteria. Specifically, we examined the association between bouted and sporadic physical activity of different intensities with the metabolic syndrome (MetS) in adults. The MetS is a common clustering of cardiometabolic risk factors (i.e., high waist circumference, triglycerides, blood pressure and glucose, low HDL-cholesterol) that is associated with an increased risk of type 2

**Table 1 Names, definitions, and examples of different physical activity patterns.**

| Name of physical activity pattern | Definition of physical activity pattern | Examples physical activities |
|---|---|---|
| Bouted MVPA | Movement intensity of $\geq 3$ METS occurring over $\geq 10$ consecutive minutes | –Walking for exercise for 30 min. <br>–Cutting grass with a push mower for 20 min. <br>–Swimming laps for 45 min. |
| Bouted LIPA | Movement intensity of primarily 1.5–2.99 METS occurring over $\geq 10$ consecutive minutes | –Walking casually around a shopping mall for an hour. <br>–Taking 30 min to prepare a meal. <br>–Playing with dog for 15 min. |
| Sporadic MVPA | Movement intensity of $\geq 3$ METS occurring over $<10$ consecutive minutes | –5 min walk to the bus stop immediately proceeded and followed by sitting. <br>–2 min of calisthenics (e.g., sit-ups, push-ups) during a TV commercial break. <br>–1 min walk to a colleague's office immediately proceeded and followed by sitting. |
| Sporadic LIPA | Movement intensity of primarily 1.5–2.99 METS occurring over $<10$ consecutive minutes | –Cleaning up after supper for 5 min. This was immediately proceeded and followed by sitting. <br>–Taking 5 min to get ready for bed (e.g., bush teeth, go to bathroom, change clothes). This 5 min was immediately proceeded by watching TV and immediately followed by sleep. |
| Embedded MVPA (i.e., sporadic MVPA embedded with bouts of LIPA) | Movement intensity of $\geq 3$ METS occurring over $<10$ consecutive minutes and occurring within bouted LIPA (as defined above) | –Taking 2 min to take out the trash. This 2 min was proceeded by 10 min of dusting and followed by 10 min of straightening things up around the home. <br>–Playing with child for 30 min. 25 min of light intensity play with 5 min of moderate intensity play embedded within the 30 min. |

**Notes.**

MVPA, moderate-to-vigorous physical activity; METS, metabolic equivalents; LIPA, light intensity physical activity.

diabetes, cardiovascular disease, certain cancers, and all-cause mortality (*Ardern & Janssen, 2007*; *Esposito et al., 2012*).

# MATERIALS AND METHODS

## Study design and participants

Participants were from the 2003–2004 and 2005–2006 cycles of the US National Health and Nutrition Examination Survey (NHANES), a nationally representative cross-sectional survey (*Centers for Disease Control and Prevention (CDC) National Center for Health Statistics (NCHS), 2003–2006*). NHANES collects data through home interviews and physical examinations in mobile exam centers. All participants gave informed consent and NHANES was approved by the National Center for Health Statistics. Ethics approval for the secondary analysis presented in this paper was given by the Health Sciences Research Ethics Board at Queen's University (file #6006002).

The current study was limited to NHANES participants aged 20 years of age and older, non-pregnant women, and those who completed the home interview, mobile exam center visit, and had fasted for at least 8 h. This left an eligible sample of 4,903. We excluded 1,579 participants with missing or invalid physical activity accelerometer data, 885 with missing MetS data, and 465 with missing covariate data. This left a final sample of 1,974. Adjusted sample weights were created from the weighting variable provided in the NHANES dataset to account for differences in age, sex, and ethnicity between the eligible sample and the final sample.

## Physical activity

Physical activity was assessed using Actigraph AM-7164 uniaxial accelerometers (Actigraph, Ft. Pensacola, FL). These accelerometers recorded average movement intensity, measured by counts in 1-minute intervals or epochs. Participants were asked to wear the accelerometers on an elasticized belt on their right hip for 7 days (*Centers for Disease Control and Prevention*). Participants were instructed to only remove the accelerometer when sleeping or when the accelerometer would get wet (e.g., bathing or swimming). After the 7 day measurement period was completed, participants mailed the accelerometers back to the NHANES researchers. The accelerometers were then tested to ensure calibration, the data were downloaded, and implausible count values were removed prior to creating an accelerometer dataset.

Further data reduction of the accelerometer dataset was carried out by the authors based on existing protocols (*Colley, Connor Gorber & Tremblay, 2010*; *Healy et al., 2011*; *Masse et al., 2005*; *Troiano et al., 2008*). Initially, we removed non-wear periods from the accelerometer dataset and calculated the wear time for each day. Non-wear periods were defined as periods with ≥90 min of zero counts, with an allowance for 2 min of counts between 0 and 100. Next, each day was coded as valid or invalid, and all accelerometer counts for invalid days were removed from the accelerometer dataset. Days were considered valid if the participant had ≥10 h of wear time. We then removed participants with 3 or fewer valid days from the accelerometer dataset.

After removing invalid days and participants with an insufficient number of valid days from the accelerometer dataset, each minute of physical activity data in the dataset was categorized into one of four intensities based on established count per minute cut-points for the Actigraph AM-7164 accelerometer (*Healy et al., 2011*; *Troiano et al., 2008*). Specifically, values between 0 and 99 counts per minute were classified as sedentary, values between 100 and 2,019 were classified as light intensity, values between 2,020 and 5,998 were classified as moderate intensity, and values ≥5,999 were classified as vigorous intensity. In a series of steps, each value was then classified as being part of a bout or a sporadic minute, as explained in the following paragraph.

Bouted MVPA was defined in the accelerometer dataset as periods of at least 10 consecutive minutes where the accelerometer counts exceeded the moderate intensity threshold, with an allowance of 20% of the counts (e.g., 2 min for a 10 min bout) being below the threshold (*Clarke & Janssen, 2014*; *Glazer et al., 2013*). Once the 20% threshold was surpassed, the bout was stopped. The time spent in bouted MVPA was summed, a daily average was created, and this information was exported from the accelerometer dataset into the main study dataset (e.g., the dataset that contained all of the other study variables). Bouted MVPA was then removed from the accelerometer dataset. Next, bouted light intensity physical activity (LIPA) was defined in the remaining accelerometer dataset as periods of at least 10 consecutive minutes where the accelerometer counts exceeded the light intensity threshold, with an allowance of 20% of the counts being below the threshold. Once the 20% threshold was surpassed, the bout was stopped. In some cases, bouted LIPA included MVPA if the amount of MVPA did not satisfy the bouted MVPA

criteria. We calculated the minutes of embedded MVPA (i.e., MVPA embedded within bouts comprised primarily of LIPA), exported this information to the main study dataset, and then removed embedded MVPA from the accelerometer dataset. We then calculated the minutes of bouted LIPA, exported this information into the main study dataset, and then removed bouted LIPA from the accelerometer dataset. After the aforementioned steps were completed, the remaining accelerometer dataset was limited to sporadic physical activity and sedentary behaviour. Within that remaining data, the daily averages for sporadic LIPA and sporadic MVPA were calculated, and that information was exported into the main study database.

## Metabolic syndrome

Presence of the MetS was based on the 2009 Joint Interim Societies definition of having at least three of the following five criteria: high waist circumference (men $\geq$ 102 cm, women $\geq$ 88 cm), high triglycerides ($\geq$150 mg/dL), low HDL-cholesterol (men < 40 mg/dL, women < 50 mg/dL), high blood pressure (systolic $\geq$ 130 mmHg or diastolic $\geq$ 85 mmHg or hypertension medication use) and high fasting glucose ($\geq$100 mg/dL or diagnosed diabetes or insulin use or glucose medication use) (*Alberti et al., 2009*).

Waist circumferences were measured to the nearest 0.1 cm at the level of the iliac crest. Blood pressure was taken four times in a seated position using a manual sphygmomanometer. The average of all four measurements was used. Blood samples were obtained after a minimum 8-hour fast and were analyzed for triglycerides, HDL-cholesterol and glucose. Triglyceride levels were measured enzymatically using a series of coupled reactions (*Centers for Disease Control and Prevention, 2008*). HDL-cholesterol was assessed using the direct HDL assay method (*Centers for Disease Control and Prevention, 2007*). Fasting plasma glucose was assessed using a hexokinase enzymatic method (*Harris et al., 1998*). Lastly, information on medication use for hypertension and diabetes and physician diagnosed diabetes (outside of gestational diabetes) were collected via self-report during the home interview.

## Covariates

Covariates were selected based on their known association with physical activity and cardiometabolic risk factors and their availability in the NHANES database. Covariates included sex, age (20–39 years old, 40–59 years old, 60+ years old), race/ethnicity (non-Hispanic white, non-Hispanic black, Mexican American, other), smoking (non-smoker, former smoker, current smoker ), alcohol (non-drinker, light to moderate drinkers defined as 1–14 drinks/week for men and 1–7 drinks/week for women, heavy drinkers defined as $\geq$15 drinks/week for men and $\geq$ 8 drinks/week for women) (*National Institute on Alcohol Abuse and Alcoholism, 1995*), and tertiles of the poverty-to-income ratio. This is a ratio of a family's income in relation to the poverty threshold for their family size and composition (*Short, 2012*).

## Statistical analysis

Statistical analyses were conducted using SAS v9.3 (SAS Institute Inc., Cary, NC) and accounted for the clustered nature of NHANES and the adjusted sample weights. Since several of the variables were not normally distributed, medians and interquartile ranges were reported. Spearman correlations were used to assess associations between the physical activity variables. Associations between physical activity and the MetS were assessed via logistic regression. Logistic regression models were run for each physical activity variable. Initially we considered whether each physical activity variable predicted MetS without adjusting for covariates or the other physical activity variables. This was followed by multivariable models that controlled for the covariates. This in turn was followed by multivariable models that adjusted for covariates and the other physical activity variables. There was evidence of collinearity for the embedded MVPA and bouted MVPA variables and therefore we did not include them in the same multivariable model. Logistic regression findings are presented as odds ratios (OR) and their associated 95% confidence intervals (CIs) per each additional 30 min/day in physical activity.

## RESULTS

Descriptive characteristics are in Table 2. The median bouted LIPA, sporadic LIPA, embedded MVPA and bouted MVPA values were 267, 100, 16, and 1 min/day, respectively. Because 95% of the sample accumulated <5 min/day of vigorous intensity physical activity, we combined moderate and vigorous intensity activity for all analyses. Furthermore, because only 11% of the sample accumulated $\geq$5 min/day of sporadic MVPA and only 1% accumulated at $\geq$10 min/day, we did not explore this form of physical activity in the regression analyses.

All of the physical activity variables were significantly correlated with each other ($p <$ 0.0001). Sporadic LIPA was negatively correlated with embedded MVPA ($R^2 = -0.10$), bouted LIPA ($R^2 = -0.19$), and bouted MVPA ($R^2 = -0.04$). Embedded MVPA was positively correlated with bouted LIPA ($R^2 = 0.29$) and bouted MVPA ($R^2 = 0.58$). Finally, bouted LIPA was positively correlated with bouted MVPA ($R^2 = 0.04$).

Before controlling for covariates, the odds ratio (95% confidence interval) for the MetS per 30 min/day of the physical activity variables were 1.15 (1.04–1.27) for sporadic LIPA, 0.93 (0.91–0.96) for bouted LIPA, 0.38 (0.30–0.47) for embedded MVPA, and 0.29 (0.21–0.41) for bouted MVPA. The corresponding values after adjusting for the covariates were 1.11 (0.99–1.24), 0.96 (0.93–0.98), 0.43 (0.34–0.54), and 0.34 (0.25–0.47). After adjusting for the covariates and the other physical activity variables (see Table 3), sporadic LIPA was not significantly associated with the MetS ($p = 0.15$). However, bouted LIPA, embedded MVPA, and bouted MVPA were all negatively associated with the MetS ($p < 0.01$, Table 3). These associations are further illustrated in Fig. 1. The final multivariable model suggested that for every 30 min/day of physical activity, there is a corresponding 4%, 57%, and 64% reduction in the relative odds of the MetS for bouted LIPA, embedded MVPA, and bouted MVPA, respectively. The associations for embedded

**Table 2** Descriptive characteristics of the study sample.

| Variable | Total ($n = 1,974$) | Men ($n = 1,120$) | Women ($n = 854$) |
|---|---|---|---|
| **Age (%)** | | | |
| 20–39 years | 38.0 (1.5) | 39.6 (2.0) | 36.4 (1.9) |
| 40–59 years | 39.8 (1.3) | 40.0 (2.0) | 39.6 (1.6) |
| 60+ years | 22.2 (1.3) | 20.3 (1.5) | 24.0 (1.7) |
| **Race/ethnicity (%)** | | | |
| Non-Hispanic white | 71.9 (2.3) | 72.5 (2.6) | 71.3 (2.4) |
| Non-Hispanic black | 11.6 (1.4) | 10.5 (1.4) | 12.7 (1.6) |
| Mexican American | 7.6 (1.1) | 8.4 (1.3) | 6.9 (1.1) |
| Other | 8.9 (1.1) | 8.6 (1.4) | 9.1 (1.4) |
| **Smoking Status (%)** | | | |
| Non-smoker | 47.5 (1.3) | 39.7 (1.8) | 55.1 (1.7) |
| Former smoker | 28.5 (1.2) | 32.5 (1.7) | 24.6 (1.4) |
| Current smoker | 24.0 (1.4) | 27.9 (1.5) | 20.3 (1.8) |
| **Alcohol consumption (%)** | | | |
| Non-drinker | 21.5 (1.5) | 19.2(1.5) | 23.7 (2.1) |
| Light to moderate drinker | 57.5 (1.7) | 56.9 (1.7) | 58.2 (2.2) |
| Heavy drinker | 21.0 (1.6) | 23.9 (1.6) | 18.2 (1.9) |
| **Physical activity (median, min/day)** | 16.2 (5.6–31.9) | 21.0 (8.7–37.5) | 10.5 (3.5–25.4) |
| Sporadic LIPA | 99.8 (81.9–116.7) | 94.5 (77.6–113.4) | 104.2 (87.4–119.7) |
| Sporadic MVPA | 2.2 (1.2–3.8) | 2.6 (1.5–4.6) | 1.8 (0.9–2.9) |
| Embedded MVPA | 16.2 (5.6–31.9) | 21.0 (8.7–37.5) | 10.5 (3.5–25.4) |
| Bouted LIPA | 266.9 (186.5–365.2) | 281.7 (185.5–378.2) | 256.4 (187.8–349.0) |
| Bouted MVPA | 1.4 (0.0–8.6) | 2.2 (0–8.9) | 0.0 (0.0–8.1) |
| **Metabolic syndrome components** | | | |
| Metabolic syndrome (%) | 41.9 (1.7) | 42.9 (2.1) | 40.9 (2.0) |
| Waist circumference | | | |
| Median, cm | 95.9 (86.3–107.7) | 99.8 (91.0–110.1) | 92.1 (82.1–103.5) |
| High waist circumference (%) | 53.1 (1.6) | 45.6 (2.1) | 60.5 (1.9) |
| Triglycerides | | | |
| Median, mg/dL | 112.6 (77.8–167.2) | 124.1 (84.2–189.6) | 104.6 (72.2–151.2) |
| High triglycerides (%) | 31.2 (1.6) | 37.0 (2.1) | 25.5 (1.6) |
| HDL-cholesterol | | | |
| Median, mg/dL | 52.1 (43.1–63.7) | 46.8 (39.9–56.6) | 58.3 (48.8–69.6) |
| Low HDL-cholesterol (%) | 62.2 (2.2) | 58.9 (2.8) | 65.4 (2.4) |
| Blood Pressure | | | |
| Median systolic, mmHg | 119 (109–131) | 121 (113–132) | 115 (107–129) |
| Median diastolic, mmHg | 70 (63–77) | 72 (64–79) | 69 (62–76) |
| High blood pressure (%) | 35.3 (1.3) | 39.3 (1.8) | 31.4 (1.9) |
| Fasting Glucose | | | |
| Median, mg/dL | 96.7 (90.2–104.0) | 98.9 (92.6–106.3) | 94.1 (88.0–101.9) |
| High fasting glucose (%) | 38.6 (2.1) | 45.9 (2.2) | 31.5 (2.3) |

**Notes.**

Data presented as prevalence (95% confidence interval) for categorical data and median (interquartile range) for categorical variables.

LIPA, light intensity physical activity; MVPA, moderate-to-vigorous intensity physical activity.

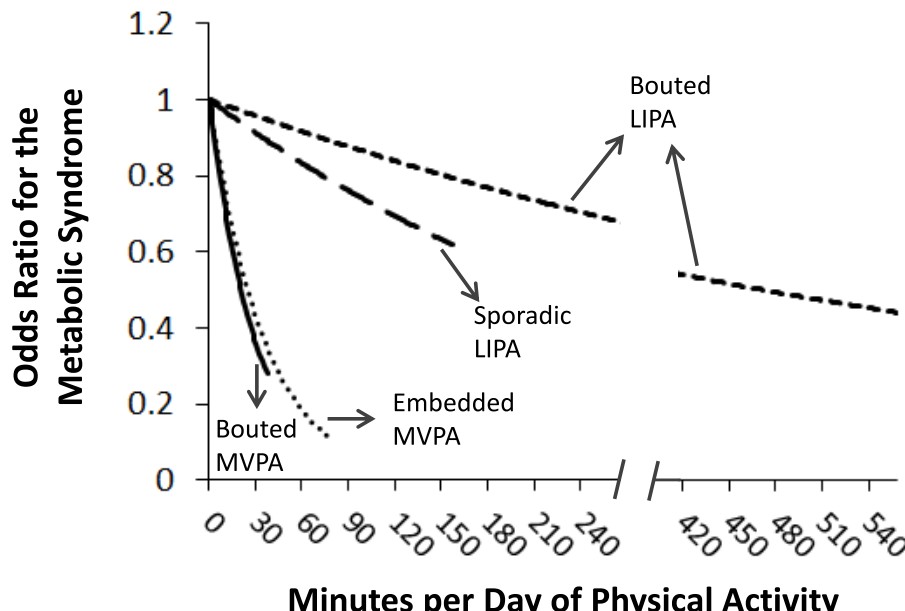

**Figure 1 Odds ratios for the metabolic syndrome per daily minute of physical activity.** Estimated odds ratios are plotted from 0 min/day to the physical activity value that corresponded to the 98th percentile in the study sample. The odds ratios for each physical activity variable are adjusted for the other physical activity variables as well as age, sex, ethnicity, poverty–income ratio, alcohol, and smoking. MVPA, moderate-to-vigorous physical activity; LIPA, light intensity physical activity.

and bouted MVPA, while not statistically different from each other, were considerably stronger than they were for bouted LIPA.

The association between the different types of physical activity and the components of the MetS are in Table 3. The statistical significance and directionality of these associations were similar to those for the MetS as a whole (exceptions: bouted LIPA was not significantly associated with low HDL-cholesterol, high blood pressure, or high glucose).

## DISCUSSION

This study examined the extent to which physical activity not meeting the criteria for bouted MVPA is associated with the MetS. A key finding was that the amount of embedded MVPA (i.e., MVPA that was embedded within bouts of primarily LIPA) was strongly associated with the MetS. Conversely, bouted LIPA and sporadic LIPA were not independently associated with the MetS. Another interesting observation was that most participants accumulated little or no truly sporadic MVPA. This type of activity accounted for only 2 min/day in the typical participant, and only 11% of participants accumulated at least 5 min/day of truly sporadic MVPA.

Observational studies, including the present study and a prior study of adults with abdominal obesity that assessed free-living physical activity over 7 days using accelerometers (*McGuire & Ross, 2012*), have found that sporadic LIPA is not independently associated with cardiometabolic risk factors. Conversely, two experimental studies observed reductions in acute postprandial glucose response when sporadic LIPA was

Robson and Janssen (2015), *PeerJ*, DOI 10.7717/peerj.1437

**Table 3** Associations between physical activity with the metabolic syndrome and the metabolic syndrome components.

| Physical activity type | Metabolic syndrome | High waist circumference | High triglycerides | Low HDL cholesterol | High blood pressure | High glucose |
|---|---|---|---|---|---|---|
| Sporadic LIPA | 0.91 (0.81–1.03)[a] | 0.88 (0.77–1.00)[a] | 0.94 (0.83–1.06)[a] | 0.99 (0.88–1.11)[a] | 0.96 (0.80–1.12)[a] | 0.99 (0.86–1.14)[a] |
| Bouted LIPA | **0.96 (0.93–0.99)**[b] | **0.96 (0.93–0.98)**[b] | **0.93 (0.90–0.96)**[b] | 0.98 (0.94–1.01)[b] | 0.98 (0.94–1.02)[b] | 0.97 (0.94–1.01)[b] |
| Embedded MVPA | **0.43 (0.33–0.55)**[a] | **0.48 (0.37–0.63)**[a] | **0.71 (0.54–0.94)**[a] | **0.79(0.66–0.95)**[a] | **0.61 (0.46–0.81)**[a] | **0.64 (0.51–0.80)**[a] |
| Bouted MVPA | **0.36 (0.26–0.49)**[b] | **0.41 (0.27–0.61)**[b] | **0.66 (0.45–0.96)**[b] | **0.73 (0.55–0.96)**[b] | **0.51 (0.35–0.74)**[b] | **0.64 (0.48–084)**[b] |

**Notes.**

Data presented as odds ratios (95% confidence interval) per 30 min/day difference in physical activity.

[a] Adjusted for age, sex, ethnicity, poverty–income ratio, alcohol, smoking, and the other physical activity variables with the exception of bouted MVPA.

[b] Adjusted for age, sex, ethnicity, poverty–income ratio, alcohol, smoking, and the other physical activity variables with the exception of embedded MVPA.

**Bold** font denotes statistically significant odds ratios ($p < 0.05$).

LIPA, light intensity physical activity; MVPA, moderate-to-vigorous physical activity.

Sporadic LIPA refers to movement intensity of primarily 1.5–2.99 metabolic equivalents (METS) occurring over <10 consecutive minutes. Bouted LIPA refers to movement intensity of primarily 1.5–2.99 METS occurring over ≥10 consecutive minutes. Embedded MVPA refers to movement intensity of ≥3 METS occurring over <10 consecutive minutes and occurring within bouted LIPA. Bouted MVPA refers to movement intensity of ≥3 METS occurring over ≥10 consecutive minutes. For examples of these different types of physical activity, refer to Table 1.

used to interrupt prolonged sedentary periods (*Bailey & Locke, 2015*; *Dunstan et al., 2012*). The discrepancy between the observational and experimental studies may reflect that the benefits of sporadic LIPA observed in the experimental studies may have been limited to an acute glucose response to insulin. These experimental studies did not find comparable changes in other cardiometabolic risk factors and did not demonstrate whether the acute changes in glucose had chronic benefits. The discrepancy between the observational and experimental studies may also reflect that the experimental studies used sporadic LIPA to break up prolonged sedentary time (e.g., 5 h), whereas the observational studies captured all sporadic LIPA irrespective of how long the sedentary period was that it interrupted. Furthermore, the accelerometer that was used in our study is imprecise at distinguishing sitting from standing and LIPA (*Kozey-Keadle et al., 2011*) and it is not able to estimate true breaks from sitting (*Lyden et al., 2014*).

Although previous studies have assessed the health benefits of non-bouted MVPA (*Clarke & Janssen, 2014*; *Glazer et al., 2013*; *Strath et al., 2008*), to our knowledge this is the first study to differentiate between truly sporadic MVPA and MVPA that was embedded within bouts of primarily LIPA. In our study the participants accumulated only 1 min of sporadic MVPA for every 8 min of embedded MVPA. Thus, the association between non-bouted MVPA and cardiometabolic health observed in previous studies is likely a reflection of embedded MVPA (*Clarke & Janssen, 2014*; *Glazer et al., 2013*; *Strath et al., 2008*). We feel that sporadic MVPA and embedded MVPA represent two distinct patterns of activity that are performed for different reasons and which may have different physiological effects. Examples of how an adult could accumulate these two patterns of activity are provided in Table 1. Sporadic MVPA could influence cardiometabolic health through the changes in lipoprotein lipase (LPL) activity associated with interrupting prolonged sedentary periods (*Hamilton, Hamilton & Zderic, 2004*). Conversely, embedded MVPA could influence cardiometabolic health through increases in catecholamine concentration and LPL activity associated with minutes of increased intensity activity during non-sedentary periods (*Greiwe et al., 1999*; *Hamilton, Hamilton & Zderic, 2004*).

Physical activity guidelines for adults recognize the health benefits of bouted MVPA but not embedded or sporadic MVPA (*Physical Activity Guidelines Advisory Committee, 2008*; *Tremblay et al., 2011*; *World Health Organization, 2010*). This reflects the evidence that was available when the guidelines were developed. This evidence was primarily based upon the results from prospective cohort studies that used self-reported instruments to measure physical activity. These instruments estimate the amount of time spent engaging in bouts of physical activity behaviors (*Troiano et al., 2014*). Many physical activity behaviors are not limited to continuous MVPA, but rather include of a combination of different movement intensities. Accelerometers, on the other hand, measure the specific time spent at different movement intensities, including both bouts and sporadic activity. Thus, self-reported instruments and accelerometers measure different constructs of physical activity and they are not interchangeable (*Troiano et al., 2014*). Over time, more studies assessing the relationship between accelerometer measured physical activity and health will

accumulate. These studies will provide a stronger and more comprehensive evidence on which to base future renditions of the physical activity guidelines.

While accelerometers represent an objective measure of physical activity, they do not capture all types of activity. Specifically, the accelerometers used in this study were uniaxial in nature and primarily captured horizontal movement at the hip. This limited their ability to accurately capture physical activity that was not step based (e.g., cycling, strength training). However, we feel that this type of activity represents the minority of leisure-time physical activity, as in a given day less than 2% of Americans engage in strength training and cycling (*Tudor-Locke, Johnson & Katzmarzyk, 2011*). The study is also limited by its cross-sectional design, precluding conclusions about the temporality of the observed associations. However, given evidence from randomized control trials and prospective cohorts that MVPA has a positive influence on cardiometabolic risk factors and the MetS (*Lakka & Laaksonen, 2007*), it may be that low MVPA was present before the MetS.

## CONCLUSION

The findings of our study suggest that MVPA does not need to occur in 10-minute bouts, as proposed in current public health guidelines for physical activity (*Haskell et al., 2007*; *Tremblay et al., 2011*; *World Health Organization, 2010*). Rather, MVPA can be accumulated during bouts of non-sedentary time, which can include bouts of primarily LIPA. Encouraging MVPA, even if just for a few minutes at a time, during prolonged bouts of LIPA may be especially relevant for individuals whose occupation involves a lot of light intensity walking or standing as there may be greater underlying potential for improvement. Future studies are needed to more clearly elucidate whether different physical activity patterns influence health. Studies that employ prospective and experimental designs are particularly warranted.

### Funding
This work was funded by a grant-in-aide from the Heart and Stroke Foundation of Ontario. The funders had no role in study design, data collection and analysis, decision to publish, or preparation of the manuscript.

### Grant Disclosures
The following grant information was disclosed by the authors:
Heart and Stroke Foundation of Ontario.

### Competing Interests
The authors declare there are no competing interests.

### Author Contributions
- Jordan Robson conceived and designed the experiments, analyzed the data, wrote the paper, prepared figures and/or tables.

- Ian Janssen conceived and designed the experiments, contributed reagents/
materials/analysis tools, reviewed drafts of the paper.

## Human Ethics

The following information was supplied relating to ethical approvals (i.e., approving body
and any reference numbers):

The National Health and Nutrition Examination Survey was approved by the National
Center for Health Statistics. Approval for the secondary analysis presented in this paper
was given by the Health Sciences Research Ethics Board at Queen's University (file
#6006002).

## Data Availability

National Health and Nutrition Examination Survey:

http://www.cdc.gov/nchs/nhanes/nhanes_questionnaires.htm.

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
