# Peer review of "Intensity of bouted and sporadic physical activity and the metabolic syndrome in adults"

_PeerJ, doi:10.7717/peerj.1437_

## Round 0.1 · original submission · Major Revisions

· Academic Editor

Major Revisions

Several major and many minor issues were raised by reviewers. Please provide a point by point response to each issue raised summarizing how each issue was addressed and where in the revised manuscript the revisions were incorporated.

Reviewer 1 ·

Basic reporting

No Comment

Experimental design

No Comment

Validity of the findings

Dear Authors,

A nice clear analysis.

Sometimes it was hard to keep track of the novel PA categories such as 'Embedded MVPA" Maybe you could give them slightly more descriptive names in Table 3.

On line 147 'calculating' should be 'calculated'.

Some people don't think much of the METs variable. Maybe Table 3 could report the associations of each of the activity categories with the 5 elements of METs separately

Additional comments

Nice work

·

Basic reporting

Figure 1 please include what these values were adjusted for in the footnote. Is it possible to add confidence intervals to figures? May need to split into 2?

All other criteria were met.

Experimental design

This paper examined the effects of different accumulation patterns of light and moderate intensity activity with metabolic syndrome. The authors tend to get caught up in definitions and measurement terminology and at times loose the focus on the big picture. This is particularly true in the first paragraph of the introduction, which immediately describes/defines the terminology before setting up the implication of the findings or the gaps that are being addressed by this paper.

Methods:
How were the covariates selected?
Were the analyses done in SUDAAN?
The description of the bivariate analysis is not clear- what are the two variables? The authors should consider presenting for each of the 4 variables on metS (sporadic, bouted LIPA, embedded and bouted MVPA) 1) after adjustment for covariates but without adjustment for other PA variables and then 2) with adjustment.

Validity of the findings

The authors place a great deal of emphasis on the finding that “sporadic MVPA” is not prevalent enough to estimate the health effects of this pattern to behavior rather than emphasizing the finding that “embedded” MVPA was associated with metS. The sentence on line 232 reads “The second major finding is that the participants accumulated an insufficient amount of truly sporadic MVPA to influence their health”. This is not entirely correct, as it is not known if people actually did accumulate sporadic MVPA whether that would affect health. The majority of the discussion is also devoted to the “distinct behavioral patterns” of embedded and sporadic MVPA- but since sporadic MVPA is essentially not found in this nationally representative population, it’s a little unclear why there is so much emphasis on this observation.

Discussion
The authors could spend more time on the fact guidelines were developed using primarily self-report questionnaires of behavior and that accelerometers are measuring something different (movement). The paper by Troiano and colleagues (BJSM, 2014) has an in depth discussion of the different constructs within the context of NHANES.

The use of accelerometers is certainly a strength of the study, but the ActiGraph is imprecise at distinguishing sitting from standing/light intensity (Kozey-Keadle et al, MSSE 2011, Kim et al., 2015 (http://dx.doi.org/10.1080/1091367X.2015.1055566). Thus, the discrepancy in findings between your and experimental studies may be that the ActiGraph is not able to estimate true “breaks” from sitting (lyden et al, MSSE 2014).

Lines 260-264 Suggest including insulin as a potential mechanism that is where the majority of evidence from experimental studies in humans seems to show benefits. Also suggest outlining some future research recommendations in this section.

The Lakka paper is an exercise training reference- should also mention future prospective studies are needed to see if “newer” patterns (non-bouted, etc) are associated with changes in risk factors for Mets.

Conclusions. First sentence should be revised to reflect what your outcome measure was “..engaging in MVPA that is not in 10-min bouts for is associated with lower odds of metabolic syndrome, suggesting…” – The second sentence seems redundant (i.e., bouts of non-sedentary which can include bouts of primarily LIPA – aren’t all non-sedentary and non-MVPA, LIPA? The last sentence of the conclusion is odd- many farming activities would be considered moderate already(https://sites.google.com/site/compendiumofphysicalactivities/Activity-Categories/lawn-garden) and it seems like a good place to discuss broader implications for general population versus very specific examples.

Additional comments

Line 270- the Tudor-Locke paper is based on a single day recall from ATUS- sentence should note this as “on a given day”- a higher proportion report engaging in strength training 2+ times/wk (older data here http://www.cdc.gov/mmwr/preview/mmwrhtml/mm5528a1.htm)


Line 133: suggest changing “initially” – sounds like it was later revised vs as a first step.
Line 147:typo “calculating”
Line 182: typo- extra )

·

Basic reporting

Standards met; clarification needed.

Introduction: Table 1
Did the authors design this table or is it from the literature? I assume the former since there is no citation.

Issues:
1. The criteria for LIPA (1.5-3 METS) is not mutually exclusive from that of MVPA (greater than or equal to 3 METS). Change MVPA to >3 METS.

2. A "5 minute walk to the bus stop" or a "1 minute walk to a colleague's office" do not meet the criteria for sporadic MVPA (>3 METS), but rather sporadic LIPA (1.5-3 METS).

Experimental design

Standards met; clarification needed.

Methods, Lines 178-179:
1-15 (men) and 1-8 (women) drinks/week is not mutually exclusive of equal to or greater than 15 and 8 drinks/week, respectively. Change heavy drinkers to >15 (men) and >8 (women) drinks/week.

Validity of the findings

Standards met; clarification needed.

Results
1. Line 198 error:
Change "99, 270, 16, and 1 minutes/day, respectively" to 267, 100, 16, and 1 minutes/day, respectively.

2. Lines 205-208:
Although all were significant, several r values were low (-.19, .21, -.32). The large sample size enhances the significance level obtained. The coefficient of determination (r squared) should be reported to show the true relationship between the variables. This will show that the low r values account for 10% or less of the variance, while the higher r values (-.44, .53, .76) are more robust (19, 28, 58% of the variance, respectively).

3. Table 3
Indicate which associations were statistically significant.

Discussion
1. Line 269
Were participants asked what types of activities they engaged in? While "less than 2% of Americans engage in strength training and cycling" was reported by Tudor-Locke et al. (2011), this cannot be assumed to be true for the sample in the current study.

2. Line 274
Change "it is likely that low MVPA was present before the MetS" to it may be that low MVPA was present before the MetS. "Likely" implies a confidence level that cannot be assumed.

Additional comments

Abstract, Line 33:
Insert "was classified" after 7 day measurement period.

Introduction, Line 88
Specify cardiometabolic risk factors (i.e. waist circumference, triglycerides, HDL, BP, Glucose).

Methods
Line 96: Add semicolon between (CDC) and National Center.
Line 147: Change "calculating" to calculated.

Discussion
Line 254: Change "associated" to association.
Line 258: Change "and adult" to an adult.

---

## Round 0.2 · accepted · Accept

· Academic Editor

Accept

The reviewers comments have now been adequately addressed.

·

Basic reporting

No comments

Experimental design

No comments

Validity of the findings

No comments

Additional comments

No comments